# Fed the same way? Exploring the influence of breastfeeding, weaning, and childhood diet on adult sex ratios through stable isotope analysis of incremental dentine in Medieval Tuscany, Italy (11th–15th c. CE)

Alessio Amaro[1]*, Antonio Fornaciari[2], Valentina Giuffra[2], Sang-Tae Kim[3], Martin Knyf[3], Paul Szpak[4], Bonnie Kahlon[1], Tracy L. Prowse[1]

1 Department of Anthropology, McMaster University, Hamilton, Canada, 2 Department of Translational Research and New Technologies in Medicine and Surgery, University of Pisa, Pisa, Italy, 3 School of Geography and Earth Sciences, McMaster University, Hamilton, Ontario, Canada, 4 Department of Anthropology, Trent University, Peterborough, Canada

* amaroa@mcmaster.ca

## Abstract

In this paper we investigate whether infant and childhood feeding practices influenced the imbalanced adult sex ratio reported in medieval Europe from historical and osteological evidence. First, we examine hypotheses for the observed imbalanced sex ratios in Europe and the evidence presented to support these hypotheses. We then use stable isotope analysis ($\delta^{13}C$ and $\delta^{15}N$) of incremental dentine in 64 first molars from adults at three medieval sites (Aulla, Badia Pozzeveri, and Montescudaio) in north-western Tuscany (11th-15th c. CE) to investigate the timing and pattern of breastfeeding and weaning. Our results show that males were exclusively breastfed for a shorter period and were weaned earlier than females in all three samples (Aulla M: 2.2 yrs, F: 2.6 yrs; Badia Pozzeveri M: 2.3 yrs, F: 3.0 yrs; Montescudaio M: 2.2 yrs, F: 2.9 yrs). In addition, both males and females consumed an isotopically similar diet early in life with greater variability during the post-weaning years. We explore how gender differences in the social roles of young males and females may have influenced infant and childhood feeding practices. It appears that these practices did not significantly disadvantage young females. In addition, we do see evidence for changing patterns of breastfeeding and weaning over time.

## Introduction

The human sex ratio (RS), an important demographic parameter, is the ratio of the number of males and females in a population. The natural male-female ratio at birth in modern human populations is estimated to be between 95 and 102 [1–3]. The normal RS range (i.e., no discrimination against either sex) is between 90 and 105.

**Data availability statement:** All relevant data are within the manuscript and its Supporting information files.

**Funding:** AA received finanancial support (Fieldwork and Conference Funding Award, Project ID: 20021030) to conduct this research from from the Department of Anthropology, School of Graduate Studies at McMaster University (https://anthropology.mcmaster.ca/) and from Mitacs (https://www.mitacs.ca/our-programs/globalink-research-award/) (Mitacs Globalink Research Award - for research abroad, Project ID: IT30649). AA, AF and TLP received financial support from the Association "Amici di San Caprasio" to conduct this research. Did the sponsors or funders play any role in the study design, data collection and analysis, decision to publish, or preparation of the manuscript? No.

**Competing interests:** The authors have declared that no competing interests exist.

The sex ratio at conception is not known, but at birth it is usually biased toward males with values around 104–107. This ratio becomes more equal during life, as males typically experience higher mortality rates [2]. The study of sex ratios in human populations has a long history, as suggested by the writings of Giovanni Villani, who collected data on the number of abandoned male and female children in medieval Florence [4], and the book written by John Graunt in 1662 on sex ratios in the city of London [5]. These initial studies were motivated by observations indicating a higher number of male newborns in the two cities. Gini [6] produced the first scientific and interdisciplinary investigation of sex ratios to study the influences of environment and society on the balance of the sexes, but it was only with Sen's [7] study that global attention was dedicated to this topic. Sen's [7] paper highlighted that more than 100 million women were missing in Asia due to disadvantageous social conditions and selective abortion. Since the 1990's, interest in sex ratio analysis has increased, with researchers investigating this demographic parameter in past populations (e.g., [8–16]). The impetus for this research was to develop explanations for the reportedly high RS in historic European populations linked to excess female mortality early in life. In a recent paper, Beltrán Tapia and Szołtysek [15] used an extensive collection of historical documents, ranging in date from 1750 to 2015 CE, to provide evidence that an imbalanced sex ratio in favour of males characterized Europe for many centuries. There is, however, currently little agreement as to what might have led to a distorted sex ratio in medieval Europe (5$^{th}$ – 15$^{th}$ c. CE), and little research has been done specifically to assess the role played by dietary differences between males and females during infancy and childhood to explain this imbalance [8–10].

Over the past decades, stable isotope analyses of skeletal human remains have been routinely used to reconstruct past diets and infant feeding practices. Nitrogen isotope values are about +3 ‰ higher due to a trophic level effect between con-sumers and their diet, first identified by DeNiro and Epstein [17] and Minagawa and Wada [18] in controlled animal feeding studies. The use of stable nitrogen isotopes to reconstruct breastfeeding, weaning and childhood dietary patterns is based on this trophic relationship and was first demonstrated by Fogel and colleagues [19] on modern mother-infant pairs. It was also noted that during exclusive breastfeeding $\delta^{13}$C values also showed an offset of about +1 ‰ [20,21]. Early studies on breast-feeding and weaning in historic European populations relied on bone samples from infants and children (e.g., [22,23]), but these represent the non-survivors, so recent research has turned to the analysis of dentine to investigate breastfeeding and weaning patterns in those individuals who survived childhood (e.g., [24,25]). The first stable isotope studies based on teeth utilized multiple samples from the same individual [26–28]; eventually dentine form one single tooth was sectioned into a few large pieces from crown to roots providing broad chronological ranges (e.g., [20]). Subsequent studies focused on obtaining a finer temporal resolution for each tooth by cutting the dentine into increasingly smaller sections. By taking advantage of the known developmental period of each tooth, researchers began to analyse a higher number of samples per tooth and assign precise ages to each dentine segment (e.g., [29–33]).

Dentine forms in conical layers, therefore, horizontal sectioning done in previous studies (e.g., [20,32]) does not yield accurate temporal resolution, as each horizontal slice encompasses dentine deposited at different developmental stages. To address this problem, Czermak and colleagues [24] developed a new method using a biopsy punch to incrementally sample dentine in a longitudinal tooth section. This new method minimised the overlap between primary dentine layers, enabling better temporal resolution for each increment. Additionally, the use of the biopsy punch method avoids areas of tooth dentine that are affected by diagenetic alterations, cavities, secondary and tertiary dentine, and cementum.

This research uses stable isotope analysis of nitrogen and carbon in incremental tooth dentine from 64 adult individuals at three sites to investigate evidence for breastfeeding and weaning patterns in medieval north-western Tuscany (11th – 15th c. CE) (Fig 1), which we hypothesize may have negatively impacted female morbidity and mortality. Previous studies using bone samples from infants and children to examine breastfeeding and weaning patterns in medieval Europe concluded that the weaning process ended between 2 and 3 years of age (e.g., [20,22,23,25,28,34–38]). However, due to methodological constraints associated with the analysis of pre-adult osteological remains, particularly the difficulty of accurately estimating biological sex, these studies do not have data on sex-based differences in infant and child feeding practices. As a result, isotopic reconstructions of breastfeeding, weaning, and infant feeding regimes in medieval Europe have thus far been unable to determine whether male and female children were subject to differential breastfeeding and weaning regimes. Furthermore, such samples reflect the feeding practices of individuals who did not survive infancy or

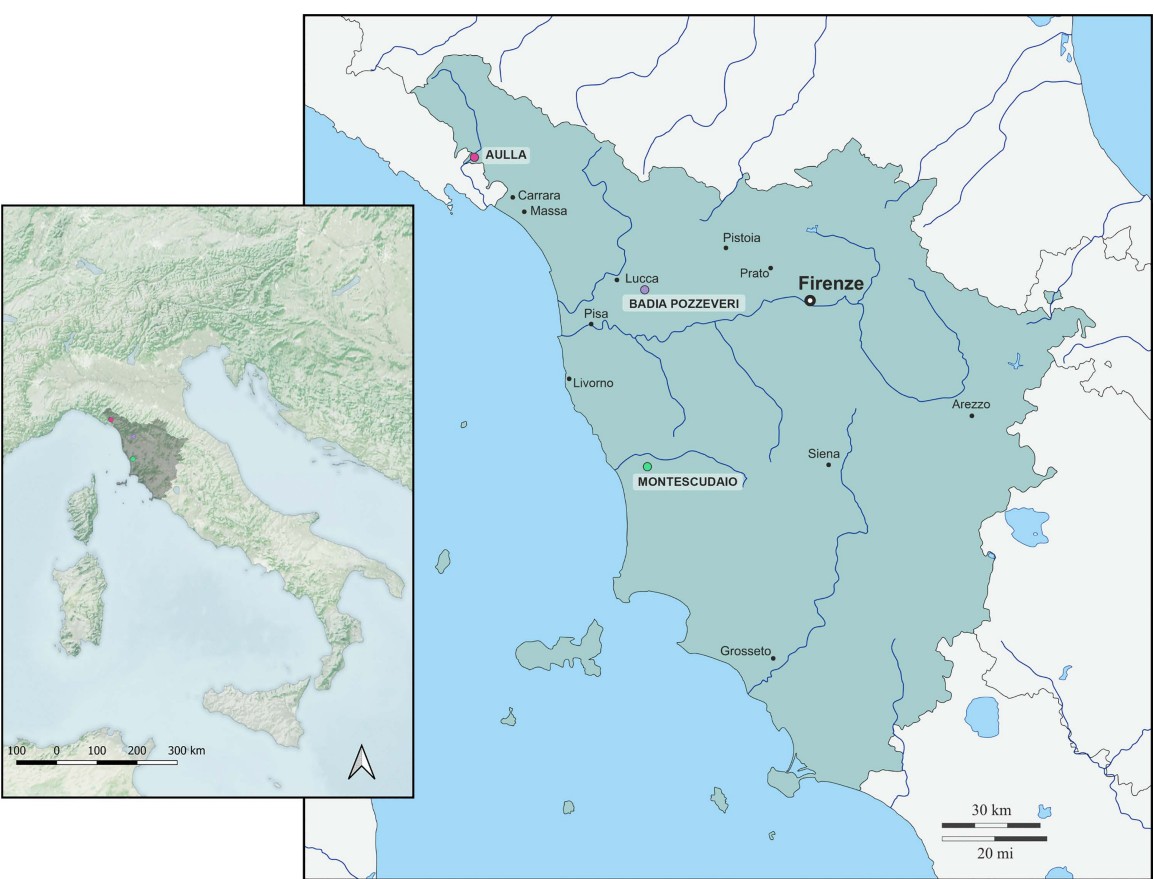

**Fig 1. Map showing the location of the sites in NW Italy. Created by Dr. Francesco Coschino, under a CC BY 4.0 license, original copyright 2025.**

early childhood. To overcome this limitation and tackle the complexities of infant/childhood feeding in medieval society and its influence on sex ratios, we targeted those individuals who survived beyond childhood to assess sex-based differences in feeding patterns.

The sites selected for this study were chosen based on their archaeological and chronological similarities and their geographical distribution in a localized area. Previous analysis of the skeletal samples from the three sites displays pronounced imbalanced sex ratios between males and females (i.e., Aulla: 183:100, Badia Pozzeveri: 236:100, Montescudaio: 200:100), comparable to the RS values found by Barbiera [8] for medieval Italy. We used the Czermak et al. [24] biopsy punch method on these molars to explore the following research questions: i) is there a difference in the timing of breastfeeding and weaning between males and females in these medieval skeletal samples?; ii) is there evidence for differences in the post-weaning diet between males and females?; and iii) does this evidence help to explain the imbalanced sex ratio observed in medieval Italy? This is the first study to explore breastfeeding, weaning and childhood diet in medieval Italy measuring carbon and nitrogen stable isotope compositions from incremental tooth dentine.

## Explanations for imbalanced sex ratios in medieval Europe

**Missing women in medieval Europe.** In recent decades, investigations into biological sex imbalances in contemporary populations demonstrated how certain socio-cultural practices, such as the unequal distribution of food, can have negative effects on population composition, reducing female survival (e.g., [7,39]). Recent research explored whether the European continent in the Modern Age (16th to 19th c. CE) was characterized by widespread gender discrimination that may have negatively impacted female survivorship and, ultimately, sex ratios (e.g., [12,14,15,40–42]). These analyses were conducted using historical documents, such as church records and population counts, on births and deaths in Europe. Similar studies were conducted for the Middle Ages (5th to 15th c. CE) (e.g., [43–45]), and what emerges from these studies is that an imbalanced sex ratio characterized Europe for many centuries. In fact, as pointed out by Beltrán and Szołtysek [15] the phenomenon of missing women existed in Europe since at least the Middle Ages, and this area may have experienced long term gender discrimination similar to the patterns observed in contemporary Asia [46].

Compared to the Modern Age in Europe, the study of this phenomenon during the Middle Ages has not received as much attention from historians due to the scarcity of written records. For example, there are limited available records to investigate the medieval dowry system to explain whether female exogamy may have played a role [47]. The few medieval documents available from Europe show a widespread imbalance between the sexes. For instance, Carolingian estate surveys (late 8th and early 9th c. CE), reported that women had a lower life expectancy and were fewer, with some accounts reporting RS ratios of 120:100 or even 140:100 in France and 112:100 in Italy [10]. Another demographic pattern deduced from these documents is the relatively greater abundance of women in medieval towns compared to rural villages [48,49]. This pattern was likely connected to better labour markets found in towns, and it is argued that urban living caused higher female mortality due to poor living conditions and later age at marriage, the latter resulting in later pregnancies and higher mortality rates during labour [50–53]. Paleodemographic reconstructions from osteological collections describe a similar scenario. In Anglo-Saxon cemeteries it was noted that females died at a younger age (e.g., [54,55]). Furthermore, the evidence suggested that women were more likely to die during their peak fertility years, probably due to complications during labour [56]. Barbiera [8] analyzed published osteological data from 28 sites in medieval Italy and found widespread imbalanced sex ratios favouring males. Barbiera [8] advanced different hypotheses to explain these missing medieval females, including: issues with biological sex attribution of skeletons, funerary segregation of females, gender-specific infanticide, and female child discrimination. Due to the limited resources available, Barbiera [8] was not able to determine which of these factors was responsible for the observed RS's in these archaeological samples and called for further study of this topic. Barbiera et al. [11] later expanded this research to include other cemeteries in Italy from a broader time range (1st-16th c. CE). The authors hypothesized that the transition from Roman to medieval society was advanced as a potential cause of the disappearance of women because parental control over daughters also

changed from a resource for the family in the Roman period to a liability (dowry system) in the medieval system. However, Barbiera and colleagues [10] acknowledged the limited historical resources available for the medieval period and the necessity for a greater effort to study the topic.

### Evidence of female infanticide

When researchers looked for potential reasons that could have caused an imbalanced sex ratio in medieval Europe, infanticide was presented as a possible explanation. For example, Coleman [57] proposed the idea of infanticide as the primary cause for the underrepresentation of women in medieval Europe through her study of polyptychs from Saint Germain-des-Prés (France). These tax censuses are one of the few written records that can be used to investigate population structure in the Early Middle Ages. These documents, estimated to be written between 801–829 CE by the monks of the monastery of Saint Germain, were produced to record information of economic importance for the monastery, including the estates belonging to the monastery and all the people who lived there. People were recorded in family groups (manse), and the head of the family, wife and children were reported. Children without parents were also recorded. Due to the detailed demographic and economic information reported, such as family size, type and amount of land worked, these documents were thought to be a detailed record of that society, which led Coleman [57] to hypothesize that infanticide was responsible for the imbalanced sex ratio in adults (from 110.3 to 259 men for every 100 women) and children (from 115.7 to 156.2 boys for every 100 girls).

Other studies used written documents (e.g., [58,59]), but none of them provided supporting evidence for infanticide. In fact, the use of historical documents to support infanticide as a leading cause for an imbalanced sex ratio was heavily criticized by Ring [60] and Siegfried [61], who both argued that polyptychs were unreliable sources to assess gender imbalances in medieval society due to their biased nature. For example, Ring [60] used the polyptych of Farfa (9th c. CE, Italy) to show how a remarkably skewed sex ratio (136.3:100) was recorded for first born children, whereas for second born children it was within normal parameters (103.3:100). These data led to the conclusion that first-born females were under recorded for economic reasons and not because of infanticide.

Consequently, there is little evidence to support the hypothesis of female infanticide. In medieval texts, infanticide was a condemned practice, as it was against Church's ideology of procreation [45,57,62,63]. Further evidence against infanticide as a common practice in medieval Europe was the practice of oblation, started by the Catholic church in the early 4th c. CE [11]. With this practice, parents were encouraged to leave their children at the door of religious buildings, where they would be cared for if the parents could no longer care for them.

### Differential treatment of female children

Here we define differential treatment as a form of systematic discrimination perpetuated by a society that routinely undervalued females. This discrimination was implemented through actions of differential parental care between males and females in early and adult life, as well as a socio-cultural milieu that considered females less valuable than males. Medieval documents describe possible sources of pervasive gender discrimination against women, such as receiving less food due to a generalised belief that females had fewer nutritional needs than men [10]. This culturally limited access to nutrients was investigated by Bullough [44], and Bullough and Campbell [64], who hypothesized that the skewed sex ratio occurred due to a limited supply of iron in women's diets, thus leading to increased mortality. It is possible that beliefs about the lower nutritional needs of women led to a series of discriminatory practices that negatively affected the number of women on the continent. In the past, the dominant view on the topic was based on Ariès' [65] research on childhood in the Middle Ages. According to Ariès [65], the concept of childhood as we know it today did not exist during the Middle Ages, and the idea of loss associated with infant death was not prevalent. Later scholars challenged this idea, such as Hanawalt [50] who used coroner's records to show the grief of parents who lost their children, both male and female, in unfortunate circumstances. Gender-based discrimination is also at the core of explanations for missing females in

historical Spain (e.g., [12,14,15,38]) and contemporary countries (e.g., [41,42,46,66]). At present, it is not clear how males were treated differently in medieval society (e.g., duration of breastfeeding, better access to food and medical care), but the reported imbalanced sex ratios suggest that males had some form of advantage that increased the likelihood of their survival to adulthood.

## Breastfeeding and weaning in the Medieval period

Breastfeeding has a central role in the development and wellbeing of a child. Breastmilk promotes the health of the baby through, for example, the transfer of antibodies [67–69] and improves the gastrointestinal environment through the introduction of beneficial bacteria into the gut microbiome [70,71]. Exclusive breastfeeding is recommended until six months of age, and the importance of breast milk is widely known, as are the potential risks of complementary food administered before the sixth month of life [72,73]. After six months, breastmilk does not provide enough nutrients to the child and other sources of nutrition must be introduced into the diet. Complementary foods may represent a danger to the health of the child since they can contain pathogens and simultaneously the beneficial effects of breastmilk are thought to diminish [74]. For these reasons, breastfeeding and weaning practices are useful indicators for demographic studies in both ancient and modern populations [75]. The first 1000 days after conception are now thought to be crucial due to the repercussions for the future development of the individual (e.g., stressful environment during the gestational period, transition from exclusively breastfeeding to weaning) [76–78].

As part of the silent majority, children received relatively little attention from medieval chroniclers and very little is known about the care of infants in medieval Europe, but the limited medieval descriptions of infant care indicate that males were breastfed longer than females [79]. Orme [80] reported that medieval children were usually breastfed by their mothers or given to a wet nurse if the family was rich. Among poorer members of society, the child could not have been breastfed at all if the mother died during labour and the family could not afford a wet nurse. This situation most likely resulted in the death of the child. Since most medieval documents only provide information about a small fraction of the society (e.g., nobles, clergy, saints, popes, kings), little information can be gathered about non-elite children and the ways they were nurtured. Medieval chroniclers promoted the idea that women should consume less food because of their lesser physical needs [81] and reported that female children were abandoned more frequently than males [10]. The duration of weaning reportedly varied between one and three years, yet similarities or differences between males and females are unknown. The Catholic Church had a positive view of breastfeeding and condemned mothers who did not breastfeed their babies for promoting their own well-being above that of their children [82]. Throughout most of the Middle Ages, paediatrics did not exist as an independent discipline but rather functioned as a branch of general medical studies, primarily driven by the aristocratic demand for healthy heirs [82]. With time, physicians promoted the use of wet nurses, which were commonly used by noble families and later became a custom even within other lower income groups [79,83,84]. Western and Arabic physicians considered breastfeeding as a continuation of the gestational period, where breastmilk was made of the same substance that fed the baby while in utero [82,85,86]. During these centuries, breastmilk was considered the best source of nutrition for the new baby, better than animal milk or that coming from a wet nurse [82]. Medieval doctors recommended that the new mother should not breastfeed her baby immediately after birth because of the traumatic experience she underwent during labour. This experience was thought to alter the balance of breastmilk, and it was suggested that she wait until balance was restored [82]. To eliminate stagnant milk and prevent blockages, it was recommended that an older woman from a lower socio-economic background suckle the new mother's breasts, or the mother was instructed to breastfeed foundlings in hospitals, thereby supplying valuable colostrum to infants from disadvantaged social classes [82]. From historical records it is known that during the 15th c. CE in France, mothers were advised to wait at least one month to breastfeed their child; in the meantime, the newborn was fed with a honey-based complementary food or was breastfed by a wet nurse [82].

Human osteological remains can provide evidence of differential access to food in past societies [87,88]. The use of stable isotopes in bone and dentine collagen allows researchers to investigate breastfeeding, weaning, and child feeding

patterns of past populations; however, few studies have focused on the Middle Ages in Europe. Most of the research that investigated infant feeding practices in medieval Europe used bone collagen, primarily from ribs (e.g., [22,23,34,36]), with few exceptions that focussed on dentine (e.g., [25]) or a combination of both (e.g., [28,37]). Isotopic research by Haydock et al. [34] and Morrone et al. [28] on medieval skeletal samples identified the end of weaning between two and three years of age but found no differences in the pattern and timing of weaning between males and females. The use of bone collagen allows a cross-sectional analysis of the sample under study, but not a longitudinal analysis of an individual's life. In addition, bone collagen studies necessarily analyse the remains of infants and children who did not survive the weaning process or only lived a short time afterwards. Dentine samples, in contrast, allow personal life histories to be traced over the period of tooth development and their study is not limited to the use of pre-adult remains. However, this approach has some limitations, including its restriction to dental remains, which precludes the reconstruction of life histories beyond the developmental period of teeth, its destructive nature, and its relatively high cost. Further, different preparation steps involved in the preparation of dentine collagen samples (e.g., filtration and solubilization steps) can impact yield and viable results in poorly preserved specimens, so a minimum amount of dentine is needed, particularly when using the biopsy punch methods as used in this study [31,89].

## Materials and methods

The three sites analyzed in this contribution were selected based on their geographical proximity in northwestern Tuscany, their similar chronological coverage and a similar chronology and archaeological context.

Aulla is located in *Lunigiana*, a historical region in northwestern Tuscany (Fig 1). The site, strategically located at the confluence of the Aulella and Magra rivers, was chosen by Marquis Adalbert I of Tuscany for the foundation of a Benedictine abbey in 884 CE. The abbey of San Caprasio gained importance in later centuries because of its proximity to the Via Francigena, an important pilgrimage route connecting northern Europe with Rome. The abbot of San Caprasio formed an independent ecclesiastical lordship in the 12th and 13th c. CE. During the 14th-15th c. CE, a period of decline began for the abbey, which was reduced to a "commendatory" abbey under the control of the Malaspina nobles and other families. The abbey survived as a commendatory abbey until its suppression in the Napoleonic era (1799–1815 CE) [90]. Archaeological excavations of the San Caprasio site were conducted by the ISCUM (Istituto di Storia della Cultura Materiale, Genoa) and the SABAP LU (Soprintendenza, Archeologia, Belle Arti e Paesaggio per le Province di Lucca e Massa Carrara) between 2001 and 2021. The excavations provided an understanding of the evolution of the abbey and monastery, as well as documenting approximately 100 burials dated between the 8th and 16th c. CE [91–93]. The archaeological context suggests that the skeletal remains associated with the abbey most likely belonged to laymen, particularly in the area outside the apses. In this study, teeth from burials belonging to the 11th-14th c. CE were sampled (n = 11).

Badia Pozzeveri is a village in Tuscany, near the city of Lucca (Fig 1). During the Middle Ages, the site was located between Lake Sesto and the Via Francigena, which connected northern Europe with Rome. Pozzeveri was first mentioned as a village in 952 CE [89–94], and in 1039 CE, the church of San Pietro di Pozzeveri was first mentioned. In 1056 CE, the church became the seat of a priests' rectory. In the late 11th c. CE, the Porcaresi, an influential local aristocratic family, invested in the creation and endowment of the monastery, which became the abbey of choice for this noble family. In 1103 CE, the monks of the Camaldolese Order took possession of the religious building, and during the 12th and 13th c. CE, the monastery increased its wealth by controlling numerous fields in the surrounding area. The abbey began to decline during the 14th c. due to military conflicts between Lucca and other surrounding cities, particularly Pisa and Florence. In 1408 CE, Pope Gregory XII sanctioned the suppression of the monastery [91–94]. The multiple archaeological campaigns conducted by the Division of Paleopathology of the University of Pisa from 2011 to 2023 unearthed more than 300 individuals, of which 100 of which date to the Middle Ages (10th-14th c. CE). The cemetery area North and West of the church were used in both the canonical and abbey phases for lay burials [95–97]. Teeth from individuals dated between the 11th and 13th c. CE were used in this study (n = 22).

Santa Maria di Montescudaio is an archaeological site located 70 km south of Pisa, in a hilly area about 10 km from the Tyrrhenian Sea (Fig 1). The church of Santa Maria, included in the diocese of Volterra, was first mentioned in 1004 CE, and its foundation probably dates to the previous century [98]. In 1091 CE it became the church of the female Benedictine abbey and was endowed with extensive land holdings in the vicinity and the proceeds of several oil mills owned by the della Gherardesca family. The cemetery was used by both religious and lay people. Over the centuries, the cloister grew considerably, reaching its greatest expansion in the 13th c. CE. Decline of the church began during the 14th c. CE with the complete abandonment of the structure in modern times [98,99]. Archaeological excavations under the supervision of the University of Pisa from 2005 to 2010 documented several cemetery phases between the late 10th and 15th-16th c. CE. [100,101] and more than 500 burials were detected. The teeth sampled in this study belong to burials dated between the 11th-15th c. CE (n = 31).

All necessary permits were obtained for the described study, which complied with all relevant regulations. The Soprin-tendenza Archeologica, Belle Arti e Paesaggio per le Province di Lucca and Massa Carrara gave permission to conduct the analysis. The Division of Paleopathology, University of Pisa, who has the samples used in this study under its control, gave permission to sample and transport the specimens used in this study. The McMaster Research Ethics Board (MREB) gave written permission (MREB #5959 – Fed the same way? Exploring the influence of breastfeeding, weaning, and childhood diet on the sex ratio of Medieval Tuscany, Italy) to conduct the analysis on archaeological human samples. The osteological samples were collected on 04/07/2022.

Each tooth selected for this study was photographed, measured and dental calculus was manually removed when pres-ent. First permanent molars were selected from 64 adult (>18 yrs) biological male (n = 39) and female (n = 25) individuals from Aulla (males = 7, females = 4), Badia Pozzeveri (males = 16, females = 6) and Montescudaio (males = 18, females = 13) to investigate breastfeeding, weaning and childhood diet patterns from dentine collagen samples. First permanent molars were selected for analysis because their dentine starts forming before birth, and is completely formed by 10 years, which likely covers the period of exclusive breastfeeding and the weaning process, as well as the post-weaning diet. For each individual, biological sex and age at death were estimated according to standard methodological procedures based on the morphological characteristics of the cranium and *os coxae*, whereas age-at-death was estimated from a combination of dental wear and the appearance of the auricular surface and pubic symphysis [102–105] (S2 File, Table 1).

Enamel from all teeth was removed using a Dremel drill and retained for future analysis. Each tooth was cleaned in distilled water in an ultrasonic bath, dried, and embedded in Buehler EpoThin epoxy. Once the epoxy was set, each tooth was then sectioned along the mesio-distal plane using a Buehler IsoMet1000 low-speed saw with a 100-rpm water-cooled diamond blade. Half of the embedded tooth section was subjected to a 100% acetone bath for 24 hours to remove the embedding material. The use of acetone does not influence the stable isotope compositions of the sample [106]. The remaining sample was rinsed, dried, and demineralized using 0.25 M hydrochloric acid (HCl) at 4° C to remove the min-eral phase, and the acid was changed every two days until demineralization was complete. Once demineralized, the pseu-domorphs were rinsed with distilled water until the pH was neutral and sampled sequentially using a 1.5 mm biopsy punch (Ted Pella, Inc) from dentine horn to root apex avoiding the pulp chamber, secondary dentine, and cementum, following the biopsy method of Czermak and colleagues [24]. Age estimation of the dentine sections was completed by comparing the samples' anatomical location to the stage of dental development following The London Atlas, including mean ages and standard deviations [22,107]. Each dentine increment was transferred into sequentially labelled glass vials, freeze-dried, weighed, and loaded into tin capsules for mass spectrometry.

A total of 699 dentine samples were analyzed. Of these, 38 samples were analyzed in a preliminary study using a Costech Elemental Analyzer with a Thermo Finnigan DELTA plus XP continuous flow-isotope ratio mass spectrometer (CF-IRMS) in the McMaster Research Group for Stable Isotopologues (MRSI) at McMaster University. The remaining 661 samples were sent to the Trent Environmental Archaeology Laboratory (TEAL) at Trent University and run on a EuroVec-tor 3000 EA and Nu Horizon mass spectrometer. Analytical accuracy was obtained through repeated measurements of

internal standards and reference materials (RMs). Routine analysis of the internal standards and RMs ensured analytical accuracy. The results of the isotopic analyses are presented as the ratio of the heavier to the lighter stable isotope (e.g., $^{13}C/^{12}C$, $^{15}N/^{14}N$) (S1 File) and are reported using a $\delta$ notation in parts per thousand or per mil (‰) (relative to VPDB and AIR respectively). Based on replicate measurements of internal standards and RMs, the analytical error at TEAL was determined to be 0.16 ‰ for $\delta^{13}C$ and 0.34 ‰ for $\delta^{15}N$. To account for linearity effects, an internal collagen standard (SRM-26, marine collagen) was analyzed in each session at variable masses to replicate the range in N contents for the samples (8–120 µg N). Six duplicates of samples run at McMaster were reanalyzed at Trent University to ensure comparability of the data between the two labs, and 50 individuals had one sample analyzed in duplicate.

Collagen preservation was checked according to %C, %N, and an atomic C:N ratio between 2.9 and 3.7 following Guiry and Szpak [108] (S1 File). Given that humic contamination, the primary factor creating inaccurate $\delta^{13}C$ values in samples of this age, would produce $\delta^{13}C$ values in the range of $C_3$ plants, C:N ratios slightly above 3.6 likely reflect samples with minimal impact on the $\delta^{13}C$ values [108]. Humic contaminants are the most likely source of the higher C:N ratios, supported by the samples with the highest C:N ratios also having the lowest $\delta^{13}C$ values. Because these humics are relatively nitrogen poor and carbon rich, the samples that have C:N ratios in the 3.5–3.7 range should have $\delta^{13}C$ and $\delta^{15}N$ values that are minimally altered [108]. Most of these samples had $\delta^{13}C$ values between −19.4 ‰ and −19.1 ‰ (lower and upper bounds at 95% range), suggesting a predominantly $C_3$-derived diet. Based on these criteria, our sample size was reduced to 508.

Statistical analysis was performed using IBM SPSS software 28.0.1.0 (142). Shapiro–Wilk tests were used to assess whether the data were normally distributed. For two sample comparison, Student's t-tests and Mann-Whitney $U$ tests were used based on the distribution of the samples. In addition, General Linear Models (repeated measure) (GLM) were used to assess variations in mean dentine values for $\delta^{13}C$ and $\delta^{15}N$ grouped by age category and biological sex. A significance level of $p < 0.05$ was used for all tests.

## Results

Each incremental dentine collagen sample was obtained using a 1.5 mm biopsy punch, which represents approximately 9 months of life (average number of subsamples 10, maximum number 14, minimum number 5). Dentine sample age categories (i.e., 0–2.9, 3.0–4.9, 5.0–6.9, 7.0–9.9) are based on those used by Czermak et al. [24]. Sequential $\delta^{13}C$ and $\delta^{15}N$ values for each individual are presented in Supplementary Material 1. Graphs of the $\delta^{13}C$ and $\delta^{15}N$ data by age, for each site, are presented in Fig 2, Fig 3, Fig 4.

### Variation based on time period

At Aulla, stratigraphic analysis identified two distinct chronological phases: phase 1 (11th–13th century CE) and phase 2 (13th–14th century CE). Statistically significant differences were observed in both $\delta^{13}C$ (t-test, $p = 0.002$) and $\delta^{15}N$ values (Mann-Whitney $U$, p = 0.008) across the two chronological phases (all sex and age data combined). To identify the source of this variation in the sample, General Linear Model (GLM) analyses of our 4 age categories, separated by sex, revealed a statistically significant difference in $\delta^{15}N$ values ($p = 0.021$) between males from phase 1 and phase 2 only within the 0–2.9 year age category (italicized in Table 1). All other comparisons were not statistically significant. Within each chronological phase, General Linear Model (GLM) analyses did not indicate any statistically significant differences in $\delta^{13}C$ or $\delta^{15}N$ values for each age category between males and females.

At Badia Pozzeveri, individuals were stratigraphically assigned to phase 1 (11th–12th century CE) and phase 2 (12th–13th century CE) based on their stratigraphic distribution. With all age categories and sexes combined, we found a statistically significant difference in $\delta^{13}C$ values between the two phases (t-test, $p = 0.016$), whereas $\delta^{15}N$ values did not show a significant difference. GLM analyses by age category and sex revealed a statistically significant difference in $\delta^{13}C$ values

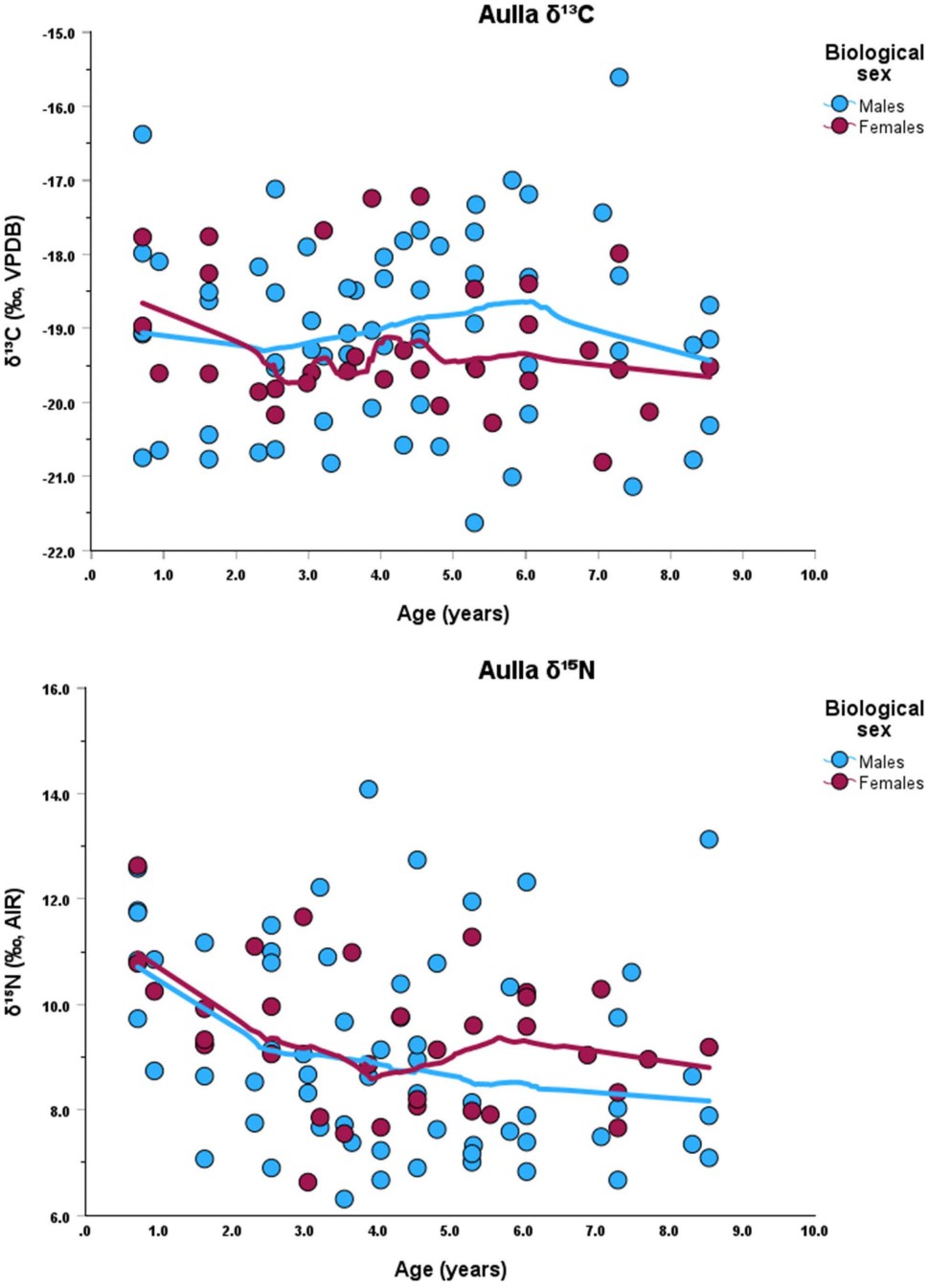

**Fig 2. Stable isotope data Lowess lines for $\delta^{13}C$ (top) and $\delta^{15}N$ (bottom) at Aulla.**

between females from phases 1 and 2 in the 5.0–6.9 year age range (p = 0.024), with individuals from phase 2 exhibiting more positive $\delta^{13}C$ values (italicized in Table 2).

Within each chronological phase, GLM analyses identified a statistically significant difference in $\delta^{13}C$ values between males and females in phase 1 within the 0–2.9 year age group (p = 0.027); a similar difference was observed for phase 2 in the same age group (t-test, p = 0.048) (bolded in Table 2). The $\delta^{13}C$ values of males and females at Badia Pozzeveri

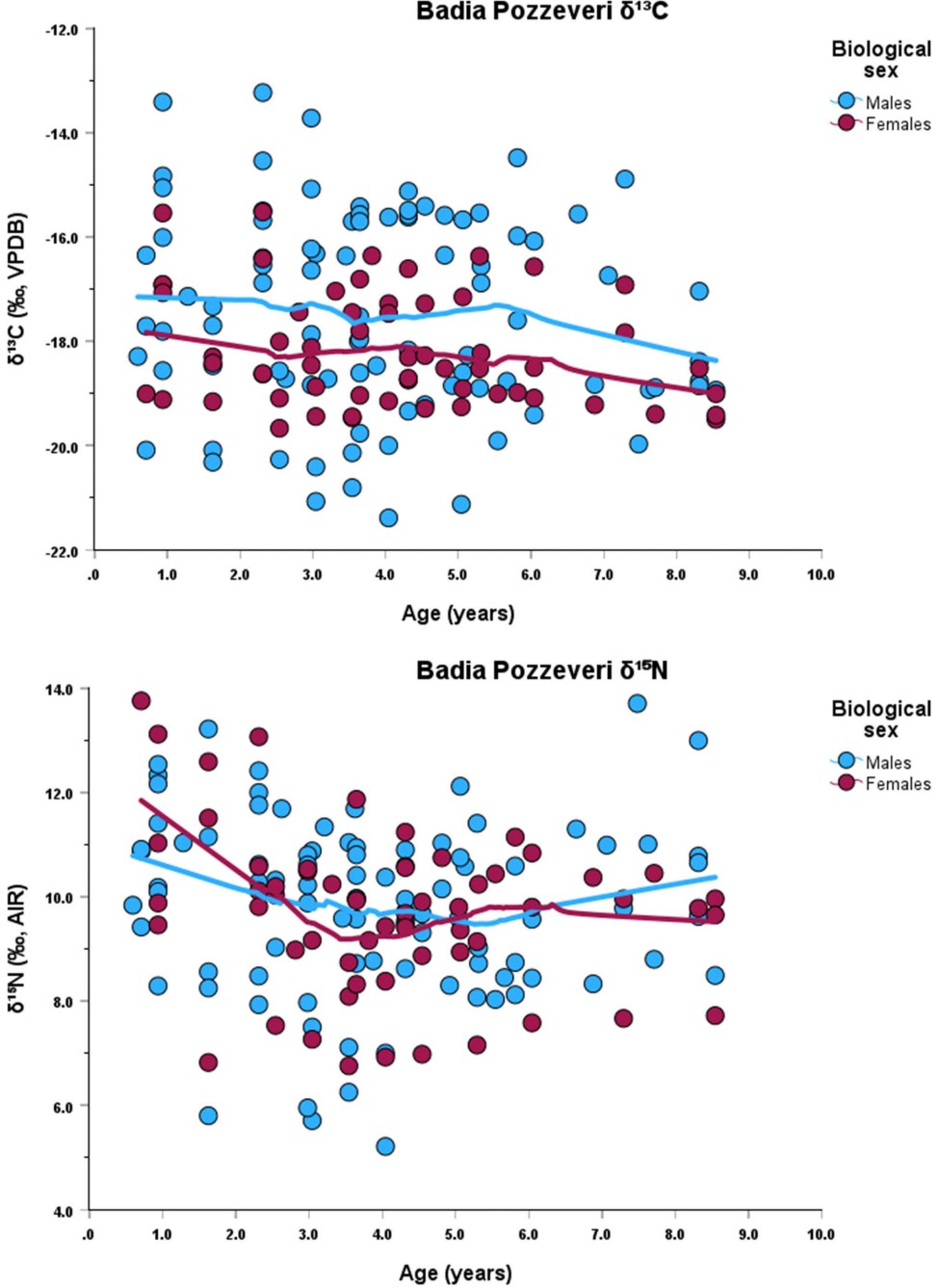

**Fig 3. Stable isotope data and Lowess lines for δ¹³C (top) and δ¹⁵N (bottom) at Badia Pozzeveri.**

(0–2.9 yrs) are highlighted because, despite identical mean values, statistical tests reveal significant sex-based differences likely due to variations in sample size and or data distribution (M = 19, F = 4).

At Montescudaio, stratigraphy enabled the distinction of three phases: phase 1 (11th century CE), phase 2 (12th–14th century CE), and phase 3 (15th century CE). T-test results showed statistically significant differences in δ¹⁵N values

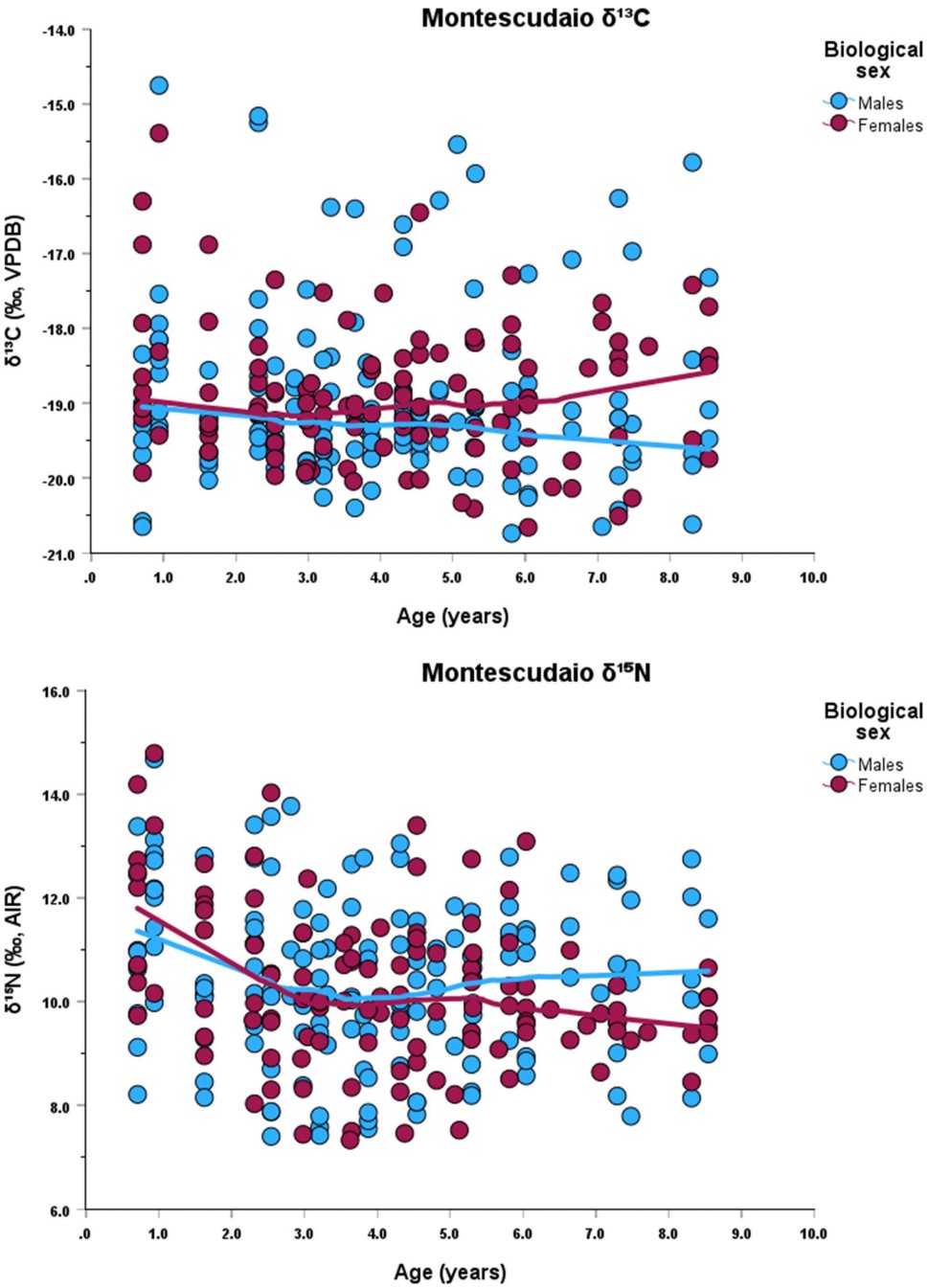

**Fig 4. Stable isotope data and Lowess lines for $\delta^{13}C$ (top) and $\delta^{15}N$ (bottom) at Montescudaio.**

between phases 1 and 2 ($p = 0.022$), and in $\delta^{13}C$ values between phases 1 and 3 ($p = 0.026$) (all age categories and sex combined). GLM analyses detected significant differences in $\delta^{15}N$ values between males in the 7.0–9.9 yrs age category between phase 1 and phase 2 ($p = 0.022$), and between phase 1 and phase 3 ($p = 0.009$) (italicized in Table 3). GLM analyses also detected a statistically significant difference between males from phase 1 and phase 2 ($p = 0.053$) and between

**Table 1. Mean δ¹³C and δ¹⁵N for each age group at Aulla by phase (phase 1 females 7.0-9.9 yrs contain only one individual). Significant differences between sexes are bolded and within the same sex across chronological phases are *italicized*.**

| Dentine section age category [after 23] | n = number of dentine sections | δ¹³C (‰) VPDB (S.D.) | δ¹⁵N (‰) AIR (S.D) | n = number of dentine sections | δ¹³C (‰) VPDB (S.D.) | δ¹⁵N (‰) AIR (S.D.) |
|---|---|---|---|---|---|---|
| *Aulla phase 1 (11ᵗʰ-13ᵗʰ c. CE)* | *Males* | | | *Females* | | |
| 0-2.9 yrs | 7 | −19.7 ± 1.0 | *9.0 ± 1.9* | 5 | −19.0 ± 0.9 | 9.8 ± 0.7 |
| 3.0-4.9 yrs | 9 | −19.5 ± 0.7 | 8.2 ± 1.5 | 7 | −18.7 ± 1.2 | 7.8 ± 0.7 |
| 5.0-6.9 yrs | 4 | −20.0 ± 1.0 | 8.7 ± 2.1 | 4 | −19.0 ± 0.7 | 9.8 ± 1.4 |
| 7.0-9.9 yrs | 4 | −20.1 ± 1.0 | 8.3 ± 1.8 | 2 | −19.5 ± 0.03 | 8.4 ± 1.1 |
| *Aulla phase 2 (13ᵗʰ-14ᵗʰ c. CE)* | *Males* | | | *Females* | | |
| 0-2.9 yrs | 9 | −18.8 ± 1.6 | *10.7 ± 1.0* | 4 | −18.9 ± 1.0 | 11.0 ± 1.2 |
| 3.0-4.9 yrs | 14 | −18.7 ± 0.9 | 9.5 ± 2.1 | 3 | −19.5 ± 0.2 | 10.8 ± 1.0 |
| 5.0-6.9 yrs | 8 | −18.0 ± 1.0 | 8.2 ± 1.7 | 4 | −19.7 ± 0.6 | 9.2 ± 1.0 |
| 7.0-9.9 yrs | 5 | −18.1 ± 1.7 | 9.3 ± 2.3 | 2 | −20.1 ± 1.1 | 9.7 ± 0.9 |

**Table 2. Mean δ¹³C and δ¹⁵N for each age group at Badia Pozzeveri by (phase 2 5.0-6.9 yrs and 7.0-9.9 yrs contain only one individual). Significant differences between sexes are bolded and within the same sex across chronological phases are *italicized*.**

| Dentine section age category [after 23] | n = number of dentine sections | δ¹³C (‰) VPDB (S.D.) | δ¹⁵N (‰) AIR (S.D) | n = number of dentine sections | δ¹³C (‰) VPDB (S.D.) | δ¹⁵N (‰) AIR (S.D.) |
|---|---|---|---|---|---|---|
| *Badia Pozzeveri phase 1 (11ᵗʰ-12ᵗʰ c. CE)* | *Males* | | | *Females* | | |
| 0-2.9 yrs | 9 | **−17.5 ± 1.4** | 10.8 ± 2.1 | 11 | **−18.4 ± 1.1** | 10.2 ± 2.0 |
| 3.0-4.9 yrs | 16 | −18.5 ± 1.4 | 9.4 ± 2.1 | 17 | −18.3 ± 1.0 | 9.0 ± 1.3 |
| 5.0-6.9 yrs | 10 | −17.8 ± 1.5 | 9.1 ± 1.2 | 10 | *−18.6 ± 0.6* | 9.6 ± 1.4 |
| 7.0-9.9 yrs | 7 | −18.3 ± 1.2 | 9.9 ± 1.3 | 6 | −18.9 ± 0.6 | 9.3 ± 1.3 |
| *Badia Pozzeveri phase 2 (12ᵗʰ-13ᵗʰ c. CE)* | *Males* | | | *Females* | | |
| 0-2.9 yrs | 19 | **−16.8 ± 2.1** | 9.9 ± 1.7 | 4 | **−16.8 ± 1.5** | 12.0 ± 1.4 |
| 3.0-4.9 yrs | 20 | −16.9 ± 2.2 | 9.5 ± 1.2 | 3 | −17.3 ± 0.1 | 8.5 ± 0.4 |
| 5.0-6.9 yrs | 6 | −17.4 ± 2.5 | 10.2 ± 1.5 | 2 | *−16.5 ± 0.1* | 9.5 ± 0.5 |
| 7.0-9.9 yrs | 5 | 17.5 ± 2.0 | 11.4 ± 1.9 | 2 | −18.2 ± 1.8 | 9.8 ± 0.2 |

males from phase 1 and males from phase 3 ($p = 0.054$) in the 3.0–4.9 age range. Because we established our cut-off for statistically significant values at $p = 0.05$, we do not consider these results as statistically significant. The GLM analyses did not identify statistically significant differences in δ¹³C values across all three phases.

Within each chronological phase, GLM analyses revealed a statistically significant difference in δ¹⁵N values between males and females in phase 3 within the 7.0–9.9 year age range ($p = 0.035$).

We have shown that there is some variation in δ¹³C and δ¹⁵N values across the different time periods, so the data are analyzed by chronological phase. However, it is important to note that the small sample size in some chronological phases may have affected the statistical analyses, and future research will be dedicated to expanding the sample size to corroborate these results.

## Weaning initiation and completion

The first 1.5 mm biopsy increment in a first permanent molar (starting at the dentine horn) represents the period from birth to ~9 months of age. When the data from the initial biopsy punch, representing the first 9 months of life, are combined

**Table 3. Mean δ¹³C and δ¹⁵N for each age group at Montescudaio by phase. Significant differences between sexes are bolded and within the same sex across chronological phases are *italicized*.**

| Dentine section age category [after 23] | n = number of dentine sections | δ¹³C (‰) VPDB (S.D.) | δ¹⁵N (‰) AIR (S.D) | n = number of dentine sections | δ¹³C (‰) VPDB (S.D.) | δ¹⁵N (‰) AIR (S.D) |
|---|---|---|---|---|---|---|
| *Montescudaio phase 1 (11th c. CE)* | **Males** | | | **Females** | | |
| 0-2.9 yrs | 18 | −19.1 ± 0.8 | 10.4 ± 1.8 | 9 | −18.4 ± 1.2 | 10.7 ± 1.5 |
| 3.0-4.9 yrs | 19 | −19.0 ± 1.1 | 8.8 ± 1.3 | 13 | −18.9 ± 0.3 | 9.9 ± 1.2 |
| 5.0-6.9 yrs | 9 | −18.5 ± 1.5 | 9.6 ± 1.3 | 10 | −18.7 ± 0.4 | 9.8 ± 1.1 |
| 7.0-9.9 yrs | 8 | −18.6 ± 1.1 | *9.2 ± 1.1* | 8 | −18.7 ± 0.6 | 9.4 ± 0.8 |
| *Montescudaio phase 2 (12th-14th c. CE)* | **Males** | | | **Females** | | |
| 0-2.9 yrs | 20 | −18.7 ± 1.5 | 11.1 ± 1.9 | 18 | −18.7 ± 1.2 | 11.3 ± 1.9 |
| 3.0-4.9 yrs | 25 | −18.9 ± 0.9 | 10.7 ± 1.4 | 21 | −19.0 ± 0.7 | 10.2 ± 1.5 |
| 5.0-6.9 yrs | 14 | −18.8 ± 1.2 | 10.5 ± 1.1 | 11 | −19.1 ± 0.9 | 10.5 ± 1.4 |
| 7.0-9.9 yrs | 9 | −18.7 ± 1.8 | *11.0 ± 1.0* | 10 | −18.5 ± 1.1 | 9.9 ± 0.4 |
| *Montescudaio phase 3 (15th c. CE)* | **Males** | | | **Females** | | |
| 0-2.9 yrs | 5 | −19.3 ± 0.2 | 11.5 ± 0.8 | 4 | −18.7 ± 0.3 | 11.4 ± 1.8 |
| 3.0-4.9 yrs | 9 | −19.6 ± 0.5 | 11.2 ± 1.1 | 6 | −19.5 ± 0.4 | 9.5 ± 1.3 |
| 5.0-6.9 yrs | 6 | −19.8 ± 0.6 | 11.1 ± 0.6 | 5 | −19.8 ± 0.4 | 9.5 ± 2.0 |
| 7.0-9.9 yrs | 5 | −19.8 ± 0.6 | ***11.9 ± 1.0*** | 3 | −20.2 ± 0.1 | **9.5 ± 0.3** |

irrespective of chronological phase, females in all three samples exhibited higher average δ¹⁵N values in the first increment compared to males (Aulla: + 1.0 ‰, Badia Pozzeveri: + 0.6 ‰, Montescudaio: + 0.5 ‰), suggesting that males began the weaning process earlier in life than females. However, when we analyzed the samples by chronological phases, t-tests revealed no statistically significant differences between males and females within the same site and chronological phase. However, there was only 1 Aulla females biopsy increment in each phase, and only 1 female in Badia Pozzeveri phase 2, so the results must be viewed with caution.

When the data are analyzed by the first dentine section (i.e., representing 0–2.9 yrs) separated by chronological phases, females continue to exhibit higher δ¹⁵N values (S2 File, Figs 5–7), except for phase 2 at Badia Pozzeveri. This trend is visible when Lowess regression lines are plotted against age, which also show elevated δ¹⁵N values for females during the first years of life (Fig 2, Fig 3, Fig 4). These elevated δ¹⁵N values in females likely reflect a longer duration of exclusive breastfeeding and an extended weaning period.

The method proposed by Velte et al. [25] was employed to estimate the cessation of weaning in each sample. This approach relies on detecting decreases in δ¹⁵N values between adjacent dentine increments, with any reduction greater than 0.2 ‰, the threshold of analytical error, indicating weaning completion. Isotopic variations were tracked from the initial increment, and values were plotted relative to the age of tooth formation. The intersection point between the trend line and the y-axis corresponds to the estimated age at which weaning concluded. This methodology permits estimation of weaning cessation at both the individual and sample levels. We cannot estimate the initiation of weaning since the first increment of dentine in our study roughly describe the first 9 months of an individual's life, which most likely also represents the exclusive breastfeeding and initiation of weaning phases. In the present study, it is utilized to evaluate differences in weaning termination between males and females, as well as across chronological phases. We based our research on the mean development of the first permanent molar, but the data about the cessation of weaning based on early and late development of the first molar are also reported (S2 File, Figs 1, 4, 7).

Across all three sites, the estimated cessation of weaning occurred between 2.2 and 2.5 years of age (S2 File, Figs 1, 4, 7). Notably, sex-based comparisons within each site (Aulla: males = 5, females = 2; Badia Pozzeveri: males = 5, females = 3; Montescudaio: males = 12, females = 9) revealed that males generally completed weaning at a younger age than females. Specifically, at Aulla, weaning concluded at approximately 2.2 years for males and 2.7 years for females; at Badia Pozzeveri, at 2.3 years for males and 3.0 years for females; and at Montescudaio, at 2.2 years for males and 2.9 years for females (S2 File, Figs 2, 5, 8).

Given the presence of multiple chronological phases at each site, we further estimated weaning endpoints within each phase when sample sizes permitted. At Aulla, during phase 1 (11th–13th century CE), males (n = 2) ended their weaning process at 2.2 years. In phase 2 (13th–14th century CE), males (n = 3) ended weaning earlier (2.1 years) than females (n = 1), who weaned at approximately 2.9 years (S2 File, Fig 3).

At Badia Pozzeveri, phase 1 (11th–12th century CE) data indicated similar weaning ages for males (n = 1; 2.9 years) and females (n = 3; 3.0 years). For phase 2 (12th–13th century CE), available data only include males (n = 4), who weaned at approximately 1.8 years (S2 File, Fig 6).

Montescudaio, characterized by three distinct chronological phases, provided more extensive data. In phase 1 (11th century CE), both males (n = 5) and females (n = 3) ceased weaning at approximately 2.4 years. In phase 2 (12th–14th century CE), males (n = 6) were weaned at 2.1 years, while females (n = 4) did so at a later age of 3.4 years. Due to limited sample availability, phase 3 (15th century CE) did not yield sufficient data for a reliable weaning age estimate (S2 File, Figs 9, 10).

## Discussion

At this point, we return to the question of the imbalanced sex ratio in medieval Europe, and whether the absence of women in historical and archaeological records might be linked to differential feeding practices during infancy and childhood. Medieval Europe was an agrarian society, and as in contemporary agrarian societies the value of males and females may have differed [109]. Beltrán Tapia and Szołtysek [15] suggested that pervasive discriminatory practices may have played a negative role in the health of females. The nature of these practices during the Middle Ages is not known, especially during childhood, although we have some clues from historical sources. A limited number of medieval chroniclers provided recommendations to feed males more than females, based on different physical needs [81]. A few sources suggest that male infants were breastfed for longer durations and received more food, presumably due to greater nutritional demands [10,79,81]. On the other hand, historical sources also point to a society that cared about sons and daughters almost equally. For example, Hanawalt [50] used coroner's records from medieval England to show a similar pattern of grief for the loss of a son or daughter. In addition, accounts of miracles show that both boys and girls were brought by their parents before the relic of the local saint hoping for divine intervention [51]. These accounts describe a society that was equally concerned about its male and female children. The isotopic evidence presented here offers the first evidence that females at these three medieval sites were weaned later than males, for reasons that we examine below.

Our stable isotope data, derived from the first permanent molars of individuals who survived infancy and childhood, show that the $\delta^{15}$N values in the 0–2.9 yrs age category display a general pattern of lower $\delta^{15}$N values among males, suggesting that males were further along in the weaning process than females by this age. This pattern is clearly seen at Badia Pozzeveri during phase 2. Here, females show $\delta^{15}$N values 2 ‰ higher than the males, reinforcing the hypothesis of different feeding strategies for both groups. The carbon isotope profiles exhibit greater variability, reflecting site-specific differences (Figs 2, 3, 4). These data were further analyzed in relation to the chronological phases identified at each site.

Overall, in all three sites, there is a $\delta^{15}$N decrease associated with the weaning process among both males and females, but females were breastfed longer and completed weaning at least 6 months later than males at all three sites (Table 4; Figs 2, 3, 4). The results also indicate that females tend to exhibit higher $\delta^{13}$C values (Tables 1, 2, 3; Figs 2, 3, 4). However, statistical analyses revealed no significant differences in $\delta^{13}$C values between males and females, except

**Table 4. Estimated weaning end (in years) for Aulla, Badia Pozzeveri and Montescudaio.**

| Aulla | Males (n) | Females (n) |
|---|---|---|
| Aulla M/F (all phases) | 2.2 (5) | 2.7 (2) |
| Phase 1 | 2.2 (2) | - (1) |
| Phase 2 | 2.1 (3) | 2.9 (1) |
| Badia Pozzeveri | Males (n) | Females (n) |
| Badia Pozzeveri M/F (all phases) | 2.3 (5) | 3.0 (3) |
| Phase 1 | 2.9 (1) | 3.0 (3) |
| Phase 2 | 1.8 (4) | – |
| Montescudaio | Males (n) | Females (n) |
| Montescudaio M/F (all phases) | 2.2 (12) | 2.9 (9) |
| Phase 1 | 2.4 (5) | 2.4 (3) |
| Phase 2 | 2.1 (6) | 3.4 (4) |
| Phase 3 | - (1) | - (1) |

at Badia Pozzeveri in the 0–2.9 age category (for both chronological phases), and among females (5.0–6.9 yrs) between phases 1 and 2 (Table 2). Modern medical literature recommends that breast milk is exclusively consumed until six months of age, and that breast milk is the best source of nutrition for the infant until that time [72]. After six months and up to twelve months, breast milk provides only about 50% of the calories needed by the infant, and approximately 30% between one year and two years [73]. This information is usually used in isotopic reconstructions to identify periods of prolonged (i.e., > six months) exclusive breastfeeding to infer that extended breastfeeding is detrimental to the health of the child [109]. However, modern medical literature also identifies the beneficial effects of prolonged breastfeeding when accompanied by supplemental feeding [110,111]. After six months, breast milk without iron supplementation may impact iron levels in infants, but overall breast milk continues to benefit the child including reducing the risk of gastrointestinal and respiratory infections [112–115]. Prolonged breastfeeding under conditions of malnutrition and/or famine can increase the child's chances of survival [116,117]. Medieval medical sources report how breast milk was highly valued and considered the optimal source of nourishment for the baby, although some accounts report how new mothers were discouraged from breastfeeding their infants during the first month after birth, thus precluding the baby's access to the beneficial effects of colostrum [82]. In some cases, prolonged breastfeeding can signify differential access to resources driven by socio-cultural practices that favoured one sex over the other (e.g., [118]), or different socio-cultural practices that unintentionally favoured females. More specifically, our data support that males and females were treated differently in terms of parental feeding strategies, with females experiencing a longer weaning period at all three sites. Whether these patterns were intentional or unintentional is a matter of debate. Alternative explanations for these patterns may involved differences in diet or the presence of physiological stress. However, we consider our proposed hypothesis to be the most parsimonious, given that the data do not yield statistically significant p values supporting the consumption of isotopically distinct diets (only exception at Badia Pozzeveri during phase 1 between 0–2.9 yrs for $\delta^{13}$C), nor do we observe pathological evidence indicative of physiological stress as contributing factor.

Our results challenge the narrative of a patrilineal medieval society in which males received better treatment from birth. It is possible that our results shed light on cultural practices in which males were exclusively breastfed for a shorter period than females for reasons related to social roles and the desire for them to reach independence earlier than females. During the Middle Ages, males played an important role in society, and parents may have opted for shorter exclusive breastfeeding and supplementary feeding to ensure a more rapid transition to becoming a contributing member of the household. This pattern is found in contemporary patrilineal societies, where male sons receive more resources than female daughters [46,119,120]. The prolonged complementary feeding experienced by females may be the result of their

later introduction into society (e.g., work and social activities) and, consequently, a longer stay within the household and a greater amount of time spent with their mothers and/or nurturers. This hypothesis is supported by the work of Hanawalt [121], who used the register of coroners in late medieval England to investigate parents' grief over the death of their children. In this research, it was found that males were more prone to injury than females, especially between the ages of two and three years. The difference seems to be caused by the more active nature of males, who tried to imitate their fathers in their outdoor activities. In a study conducted on a 18th and 19th century population from London, females were found to have an increased $\delta^{15}N$ compared to males [122]. Given the paucity of information about infant feeding practices, the authors suggested that the difference could have been caused by dietary or physiological differences. We argue that, at least in medieval north-western Tuscany, if women disappeared it was because they were subjected to discriminatory practices later in their lives, such as adolescence and during their early reproductive years, and not in their early years.

Our stable isotope data, together with historical accounts, describe a society that gave equal access to isotopically similar food resources to males and females during their first years of life, but females had longer access to exclusive breastfeeding and extended complementary feeding, signifying the prolonged and beneficial presence of breast milk in their diet (S2 File, Figs 4–6). The post-weaning years, instead, do not reveal differences between the two sexes, except for Montescudaio during phase 3 where males and females had differential access to $\delta^{15}N$-enriched resources, with males consuming more terrestrial animal products. We also see a trend of increasing $\delta^{15}N$ values among young males (7.0–9.9 yrs) between time periods at Montescudaio, indicating a greater consumption of higher trophic level foods by boys at this age. Given the proximity of Montescudaio to the Tyrrhenian Sea, the consumption of marine products is not surprising.

## Conclusions

The debate about the ultimate causes of the disappearance of women in medieval Europe is still open. At this point, further research is needed to compare infant feeding patterns between individuals who survived to adulthood with those who did not. Our current study shows that males and females in medieval north-western Tuscany were subject to different feeding schedules that were potentially influenced by gender-based parenting strategies. The results of our stable isotope analysis present a more fine-grained analysis of breastfeeding and weaning of infants and children in medieval north-western Tuscany between the 11th c. CE and the 15th c. CE, showing that females were exclusively breastfed longer than males, and that the weaning period was extended for females, ending ~6–8 months later than their male counterparts. Our data provide new insights into infant and childhood feeding in medieval Tuscany and help to describe the gendered feeding strategies of parents. Based on this evidence we argue that the disappearance of medieval females should be attributed to discriminatory practices after infancy and childhood. Further research is underway to analyze breastfeeding and weaning patterns in a group of non-survivors from the same sites used in this contribution. This new research will shed light on the feeding patterns of individuals who did not reach adulthood.

## Supporting information

**S1 File. The file includes the raw data ($\delta^{13}C$, $\delta^{15}N$, biological sex and dentine mean estimated formation age [23])** **for all the individuals from Aulla, Badia Pozzeveri and Montescudaio.**
(DOCX)

**S2 File. The file includes a supplementary table with extra information about the individuals analysed (site, chronology, biological sex, age-at-death) and a table for each site with the number of sections, mean $\delta^{13}C$ and $\delta^{15}N$ measured from first sections.** Additionally, here are reported figures for each site and chronology representing the estimated weaning end (all individuals combined, and males and females separated).
(DOCX)

## Acknowledgments

We thank the Soprintendenza Archeologica, Belle Arti e Paesaggio per le Province di Lucca and Massa Carrara (Dr. Marta Colombo and Dr. Neva Chiarenza) for their support to the study project. We thank Dr. Enrico Giannichedda and Dr. Monica Baldassarri, who directed the excavations in Aulla (2001–2006) and Montescudaio (2005–2010) respectively. Thank-you to Dr. Francesco Coschino for his assistance with Fig 1.

## Author contributions

**Conceptualization:** Alessio Amaro, Tracy L. Prowse.

**Data curation:** Alessio Amaro.

**Formal analysis:** Alessio Amaro.

**Funding acquisition:** Alessio Amaro, Antonio Fornaciari, Tracy L. Prowse.

**Investigation:** Alessio Amaro, Sang-Tae Kim, Martin Knyf, Paul Szpak, Bonnie Kahlon.

**Methodology:** Alessio Amaro, Bonnie Kahlon.

**Project administration:** Alessio Amaro, Tracy L. Prowse.

**Resources:** Antonio Fornaciari, Valentina Giuffra.

**Supervision:** Tracy L. Prowse.

**Visualization:** Alessio Amaro.

**Writing – original draft:** Alessio Amaro, Tracy L. Prowse.

**Writing – review & editing:** Alessio Amaro, Antonio Fornaciari, Valentina Giuffra, Sang-Tae Kim, Martin Knyf, Paul Szpak, Bonnie Kahlon, Tracy L. Prowse.

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
