## [Decision Letter · Decision Letter 0]

25 Sep 2025

PONE-D-25-41106Fed the Same Way? Exploring the Influence of Breastfeeding, Weaning, and Childhood Diet on Adult Sex Ratios Through Stable Isotope Analysis of Incremental Dentine in Medieval Tuscany, Italy (11th – 15th c. CE)PLOS ONE?

Dear Dr. Amaro,

Thank you for submitting your manuscript to PLOS ONE. After careful consideration, we feel that it has merit but does not fully meet PLOS ONE’s publication criteria as it currently stands. Therefore, we invite you to submit a revised version of the manuscript that addresses the points raised during the review process.

We look forward to receiving your revised manuscript.

Kind regards,

Dario Piombino-Mascali, Ph.D.

Academic Editor

PLOS ONE

**Journal Requirements:**

1. When submitting your revision, we need you to address these additional requirements. Please ensure that your manuscript meets PLOS ONE's style requirements, including those for file naming. The PLOS ONE style templates can be found at https://journals.plos.org/plosone/s/file?id=wjVg/PLOSOne_formatting_sample_main_body.pdf and https://journals.plos.org/plosone/s/file?id=ba62/PLOSOne_formatting_sample_title_authors_affiliations.pdf 2. In your manuscript, please provide additional information regarding the specimens used in your study. Ensure that you have reported human remain specimen numbers and complete repository information, including museum name and geographic location.  If permits were required, please ensure that you have provided details for all permits that were obtained, including the full name of the issuing authority, and add the following statement: 'All necessary permits were obtained for the described study, which complied with all relevant regulations.' If no permits were required, please include the following statement: 'No permits were required for the described study, which complied with all relevant regulations.' For more information on PLOS One's requirements for paleontology and archeology research, see https://journals.plos.org/plosone/s/submission-guidelines#loc-paleontology-and-archaeology-research. 3. Thank you for stating the following financial disclosure: AA received finanancial support (Fieldwork and Conference Funding Award, Project ID: 20021030) to conduct this research from from the Department of Anthropology, School of Graduate Studies at McMaster University (https://anthropology.mcmaster.ca/) and from Mitacs (https://www.mitacs.ca/our-programs/globalink-research-award/) (Mitacs Globalink Research Award - for research abroad, Project ID: IT30649).AA, AF and TLP received financial support from the Association "Amici di San Caprasio" to conduct this research.Did the sponsors or funders play any role in the study design, data collection and analysis, decision to publish, or preparation of the manuscript? No   Please state what role the funders took in the study.  If the funders had no role, please state: "The funders had no role in study design, data collection and analysis, decision to publish, or preparation of the manuscript." If this statement is not correct you must amend it as needed. Please include this amended Role of Funder statement in your cover letter; we will change the online submission form on your behalf. 4. Thank you for stating the following in the Acknowledgments Section of your manuscript: We thank the Soprintendenza Archeologica, Belle Arti e Paesaggio per le Province di Lucca and Massa Carrara (Dr. Marta Colombo and Dr. Neva Chiarenza) for their support to the study project. We thank Dr. Enrico Giannichedda and Dr. Monica Baldassarri, who directed the excavations in Aulla (2001-2006) and Montescudaio (2005-2010) respectively. We also thank the Department of Anthropology, the School of Graduate Studies at McMaster University, Mitacs and the Association ‘Amici di San Caprasio’ for their financial support of this research. Thank-you to Dr. Francesco Coschino for his assistance with Fig 1. We note that you have provided funding information that is not currently declared in your Funding Statement. However, funding information should not appear in the Acknowledgments section or other areas of your manuscript. We will only publish funding information present in the Funding Statement section of the online submission form. Please remove any funding-related text from the manuscript and let us know how you would like to update your Funding Statement. Currently, your Funding Statement reads as follows: AA received finanancial support (Fieldwork and Conference Funding Award, Project ID: 20021030) to conduct this research from from the Department of Anthropology, School of Graduate Studies at McMaster University (https://anthropology.mcmaster.ca/) and from Mitacs (https://www.mitacs.ca/our-programs/globalink-research-award/) (Mitacs Globalink Research Award - for research abroad, Project ID: IT30649).AA, AF and TLP received financial support from the Association "Amici di San Caprasio" to conduct this research.Did the sponsors or funders play any role in the study design, data collection and analysis, decision to publish, or preparation of the manuscript? No  Please include your amended statements within your cover letter; we will change the online submission form on your behalf. 5. When completing the data availability statement of the submission form, you indicated that you will make your data available on acceptance. We strongly recommend all authors decide on a data sharing plan before acceptance, as the process can be lengthy and hold up publication timelines. Please note that, though access restrictions are acceptable now, your entire data will need to be made freely accessible if your manuscript is accepted for publication. This policy applies to all data except where public deposition would breach compliance with the protocol approved by your research ethics board. If you are unable to adhere to our open data policy, please kindly revise your statement to explain your reasoning and we will seek the editor's input on an exemption. Please be assured that, once you have provided your new statement, the assessment of your exemption will not hold up the peer review process. 6. Please include your full ethics statement in the ‘Methods’ section of your manuscript file. In your statement, please include the full name of the IRB or ethics committee who approved or waived your study, as well as whether or not you obtained informed written or verbal consent. If consent was waived for your study, please include this information in your statement as well. 7. We note that Figure 1 in your submission contain map/satellite images which may be copyrighted. All PLOS content is published under the Creative Commons Attribution License (CC BY 4.0), which means that the manuscript, images, and Supporting Information files will be freely available online, and any third party is permitted to access, download, copy, distribute, and use these materials in any way, even commercially, with proper attribution. For these reasons, we cannot publish previously copyrighted maps or satellite images created using proprietary data, such as Google software (Google Maps, Street View, and Earth). For more information, see our copyright guidelines: http://journals.plos.org/plosone/s/licenses-and-copyright. We require you to either present written permission from the copyright holder to publish these figures specifically under the CC BY 4.0 license, or remove the figures from your submission: a. You may seek permission from the original copyright holder of Figure 1 to publish the content specifically under the CC BY 4.0 license.   We recommend that you contact the original copyright holder with the Content Permission Form (http://journals.plos.org/plosone/s/file?id=7c09/content-permission-form.pdf) and the following text:“I request permission for the open-access journal PLOS ONE to publish XXX under the Creative Commons Attribution License (CCAL) CC BY 4.0 (http://creativecommons.org/licenses/by/4.0/). Please be aware that this license allows unrestricted use and distribution, even commercially, by third parties. Please reply and provide explicit written permission to publish XXX under a CC BY license and complete the attached form.” Please upload the completed Content Permission Form or other proof of granted permissions as an "Other" file with your submission. In the figure caption of the copyrighted figure, please include the following text: “Reprinted from [ref] under a CC BY license, with permission from [name of publisher], original copyright [original copyright year].” b. If you are unable to obtain permission from the original copyright holder to publish these figures under the CC BY 4.0 license or if the copyright holder’s requirements are incompatible with the CC BY 4.0 license, please either i) remove the figure or ii) supply a replacement figure that complies with the CC BY 4.0 license. Please check copyright information on all replacement figures and update the figure caption with source information. If applicable, please specify in the figure caption text when a figure is similar but not identical to the original image and is therefore for illustrative purposes only.The following resources for replacing copyrighted map figures may be helpful: USGS National Map Viewer (public domain): http://viewer.nationalmap.gov/viewer/The Gateway to Astronaut Photography of Earth (public domain): http://eol.jsc.nasa.gov/sseop/clickmap/Maps at the CIA (public domain): https://www.cia.gov/library/publications/the-world-factbook/index.html and https://www.cia.gov/library/publications/cia-maps-publications/index.htmlNASA Earth Observatory (public domain): http://earthobservatory.nasa.gov/Landsat:
http://landsat.visibleearth.nasa.gov/USGS EROS (Earth Resources Observatory and Science (EROS) Center) (public domain): http://eros.usgs.gov/#Natural Earth (public domain): http://www.naturalearthdata.com/ 8. We note that there is identifying data in the Supporting Information file. Due to the inclusion of these potentially identifying data, we have removed this file from your file inventory. Prior to sharing human research participant data, authors should consult with an ethics committee to ensure data are shared in accordance with participant consent and all applicable local laws. Data sharing should never compromise participant privacy. It is therefore not appropriate to publicly share personally identifiable data on human research participants. The following are examples of data that should not be shared: -Name, initials, physical address-Ages more specific than whole numbers-Internet protocol (IP) address-Specific dates (birth dates, death dates, examination dates, etc.)-Contact information such as phone number or email address-Location data-ID numbers that seem specific (long numbers, include initials, titled “Hospital ID”) rather than random (small numbers in numerical order) Data that are not directly identifying may also be inappropriate to share, as in combination they can become identifying. For example, data collected from a small group of participants, vulnerable populations, or private groups should not be shared if they involve indirect identifiers (such as sex, ethnicity, location, etc.) that may risk the identification of study participants. Additional guidance on preparing raw data for publication can be found in our Data Policy (https://journals.plos.org/plosone/s/data-availability#loc-human-research-participant-data-and-other-sensitive-data) and in the following article: http://www.bmj.com/content/340/bmj.c181.long. Please remove or anonymize all personal information (<specific identifying information in file to be removed>), ensure that the data shared are in accordance with participant consent, and re-upload a fully anonymized data set. Please note that spreadsheet columns with personal information must be removed and not hidden as all hidden columns will appear in the published file. 9. If the reviewer comments include a recommendation to cite specific previously published works, please review and evaluate these publications to determine whether they are relevant and should be cited. There is no requirement to cite these works unless the editor has indicated otherwise. 

**Additional Editor Comments:**

Dear Mr Amaro,

Please review the paper according to the reviewers' comment, and take this chance to double-check, for a last time, if all references have been accurately formatted according to the journal style. I look forward to read the reviewed manuscript.

Reviewers' comments:

**Comments to the Author**

1. Is the manuscript technically sound, and do the data support the conclusions?

Reviewer #1: Yes

Reviewer #2: Yes

2. Has the statistical analysis been performed appropriately and rigorously?

Reviewer #1: Yes

Reviewer #2: Yes

3. Have the authors made all data underlying the findings in their manuscript fully available?

Reviewer #1: Yes

Reviewer #2: Yes

4. Is the manuscript presented in an intelligible fashion and written in standard English?

Reviewer #1: Yes

Reviewer #2: Yes

**Reviewer #1:**  This is a really well written piece of research with solid questions focusing on infant feeding practices in medieval Tuscany and their possible effects on the sex ratio imbalance. The paper is well structured and arranged, with great historical background, meticulous referencing, as well as proper English.

However, there are some minimal aspects that I would advise to reconsider to improve the clarity of the paper. These are also reported in the pdf as comments in specific highlighted sections.

1. I feel that sometimes, during the train of thoughts, the authors tend to make logical leaps in the discourse, assuming that the reader is aware of the reasoning behind some particular statements. For instance, it is not clearly stated at the beginning for which actual reasons you decided to concentrate on the infant feeding strategies as the main cause of an imbalanced sex ratio in this specific population. Although it is indeed a possible explanation, it wasn't stated why you chose to give priority to this hypothesis rather than others such as nutritional imbalance in adolescence or adulthood, as well as labour or cultural choices. A similar dissonance occurs in the infanticide section, in which it is again unclear how the cited authors arrived to such extreme statements from solely census documents. I would advise to recheck these points, perhaps adding a couple more statements giving more clarity.

2. It is not clear from the practical point of view how you define the weaning process: how and where you see it starting and especially how you see it ending in the isotopic data. You give very specific ages for the end of the process (and not an age range), which give the impression of an abrupt moment while the actual process is quite gradual and often moves back and forth. It would be good to show these clearly in the graphs, perhaps using one as a model, and explain them very clearly in the text.

3. There can be several reasons for the raising of N values, also when considering incremental dentine sections. Although here it is probable that the rise was influenced by breastfeeding, I think that also diet, stress and pathology should be mentioned as possible concomitant factors. Are some of these individuals showing sings of childhood stress such as enamel hypoplasia on other teeth, or cribra orbitalia? It could be interesting to check this and see if the timing is influenced by this as well.

4. In the introduction, the authors correctly report the limitations of using a cross-sectional approach for studying infant feeding patterns. However, I think the paper misses an essential section reporting the limitations of incremental dentine sampling in general (which unfortunately are several), as well as the actual limitations of this specific study, namely the very few individuals of each sex that are funneled down when you pick the age categories per site. It is very briefly mentioned in the results, but I think it should also be clearly stated in a dedicated paragraph.

5. I'm not sure if this is a problem of the pdf for reviewers, but the images look really blurry. Please re-check them.

With this said, I think that with these few modifications, the paper is ready for publication.

**Reviewer #2:**  Peer review report on “Fed the Same Way? Exploring the Influence of Breastfeeding, Weaning, and Childhood Diet on Adult Sex Ratios Through Stable Isotope Analysis of Incremental Dentine in Medieval Tuscany, Italy (11th – 15th c. CE)”

1. Recommendation

Minor revision

2. Comments to Author:

Title: “Fed the Same Way? Exploring the Influence of Breastfeeding, Weaning, and

Childhood Diet on Adult Sex Ratios Through Stable Isotope Analysis of Incremental

Dentine in Medieval Tuscany, Italy (11th – 15th c. CE)”

Authors: Alessio Amaro, Antonio Fornaciari, Valentina Giuffra, Sang-Tae Kim, Martyn Knyf Paul Szpak, Bonnie Kahlon, Tracy L. Prowse

Overview and general recommendation:

Amaro et al. present a valuable study addressing two key issues: (1) the phenomenon of missing female individuals in the funerary context, and (2) differences in infant feeding and children dietary practices between male and female individuals from medieval populations in Aulla, Badia Pozzeveri, and Montescudaio, Italy. The dataset generated represents an important contribution to the field and is suitable for publication. The results are presented clearly and in organized manner; however, the manuscript would benefit from the inclusion of additional plots, such as plots of individual nitrogen and carbon isotope values by age, to improve data accessibility. The interpretation that females were breastfed for a longer duration than males across all three sites requires reconsideration, as the small sample sizes – particularly when subdivided by phase, with the exception of Montescudaio phase 2 – limit the strength of this conclusion. A more nuanced discussion of possible reasons for longer breastfeeding and weaning practices among females would strengthen the argument. As these revisions are relatively minor and can be implemented without restructuring, I recommend the manuscript for publication pending minor revisions.

Comments and suggestions:

Introduction

Page 2, lines 50-51: the transition into the paragraph of nitrogen stable isotope feels a little abrupt. The flow could be improved by adding a short introductory sentence to indicate that the following section will focus on the application of stable isotope analysis to infant feeding practices in archaeological populations.

Page 3, line 59: ‘has turned to the analysis of the sequential dentine to investigate breastfeeding and weaning patterns in those individuals who survived childhood (e.g., [23, 24]).’ I suggest being more specific here by emphasizing the use of sequential or incremental dentine analysis. While dentine as a window into childhood diet was already explored in the early 2000s through bulk dentine analysis, the latest studies apply a different approach, and it would be useful to highlight this methodological distinction.

Page 4, line 95: The phrasing ‘investigations into gender imbalances in contemporary populations’ should be reconsidered. Since sex and gender are distinct concepts, and the text later refers to ‘female survival’, it would be clearer and more accurate to frame this specifically in terms of sex rather than gender.

In Section 2.1. It would be helpful to clarify whether there is any evidence or discussion regarding female exogamy. Including this information could provide additional context for interpreting the sex-specific patterns observed in the populations.

Materials and Methods

Page 11, lines 276-286: It would be helpful to include the approximate number of individuals uncovered in the site description. Since other sites report these numbers, adding this information here would ensure consistency and enhance clarity regarding the material analyzed.

Page 11, line 293: There is an inconsistency in site names. Elsewhere in the text, as well as in plots and tables, the site is referred to as Aulla rather than San Caprasio.

Page 11, line 300: If you choose to use ‘innominate’, it would be clearer to use the more precise and formal term ‘innominate bone’. Alternatively, following current anatomical terminology, the preferred academic term is ‘os coxae’ or ‘pelvic bone’.

Results

Page 14, line 354: Could you please provide information on how many subsamples were obtained per tooth on average using the biopsy punch, as well as the maximum and the minimum number?

Page 15, lines 379¬-382: The statement here is confusing. Based on the bolded means provided in Table 2, there does not appear to be a statistically significant difference in carbon values between males and females in phase 2 within the 0-2.9 year age group or ‘males exhibiting more positive δ13C values’ – the reported means are exactly the same (-16.8). Could you please rephrase or clarify this section to make the intended point clearer?

Page 16, Table 2. Could you please comment on the nitrogen values in Badia Pozzeveri phase 2 between males and females within the 0-2.9 year age group? The difference is slightly more than 2 ‰. What are your thoughts on the potential significance or interpretation of this difference?

Page 17, lines 408-412: Would you consider other possible explanations for the higher average nitrogen values observed in females in the first increment, apart from weaning? Plots for individual carbon and nitrogen curves would be very helpful to include. They could allow readers to better assess whether any evidence of catabolic activity or potential “stress bubbles” is present in the data.

Discussion:

Page 20, lines 473-484: This paragraph would be more suitable for the Introduction section, where it could be framed as a general problem in reconstructing infant and childhood diet.

Page 21, lines 514-517: Is there any evidence of non-specific stress indicators, such as enamel hypoplasia, that could point to health restrictions during the first years of life? Such indicators might provide useful context for interpreting infant feeding practices, particularly if they were adapted to individual-specific needs.

Page 21, from 518 line: I would recommend including some additional references on dietary differences between male and female children. Although these works are not specifically focused on Medieval contexts, they could still be useful for discussing your results within a broader comparative framework.

• Henderson et al., 2014 https://doi.org/10.1002/ajpa.22554

• Henderson, 2015 (PhD thesis)

• Miller et al., 2020 https://doi.org/10.1002/ajpa.24033

Page 22, lines 543-545: Is there any adult or animal isotopic data available that could support or contextualize?

**Do you want your identity to be public for this peer review?** For information about this choice, including consent withdrawal, please see our Privacy Policy

Reviewer #1: No

Reviewer #2: No

---

## [Author Response · Author response to Decision Letter 1]

14 Nov 2025

Dear Editor,

Thank you for your feedback and for the opportunity to revise our manuscript “Fed the Same Way? Exploring the Influence of Breastfeeding, Weaning, and Childhood Diet on Adult Sex Ratios Through Stable Isotope Analysis of Incremental Dentine in Medieval Tuscany, Italy (11th – 15th c. CE)”. We greatly appreciate the reviewers’ insightful comments, which have helped us strengthen our work.

Below, we provide a detailed list of the editor and reviewers’ feedback along with our responses and the corresponding revisions made. All changes are clearly highlighted in the attached document titled “Revised Manuscript with Track Changes”.

Editor:

- We have updated our manuscript to meet PLOS ONE’s style requirements.

2. In your manuscript, please provide additional information regarding the specimens used in your study. Ensure that you have reported human remain specimen numbers and complete repository information, including museum name and geographic location.

For more information on PLOS One's requirements for paleontology and archeology research, see https://journals.plos.org/plosone/s/submission-guidelines#loc-paleontology-and-archaeology-research.

- We have provided more details on the samples used in this study and the permissions obtained for their study (lines 309-313, 316-319).

AA received finanancial support (Fieldwork and Conference Funding Award, Project ID: 20021030) to conduct this research from from the Department of Anthropology, School of Graduate Studies at McMaster University (https://anthropology.mcmaster.ca/) and from Mitacs (https://www.mitacs.ca/our-programs/globalink-research-award/) (Mitacs Globalink Research Award - for research abroad, Project ID: IT30649).

AA, AF and TLP received financial support from the Association "Amici di San Caprasio" to conduct this research.

Did the sponsors or funders play any role in the study design, data collection and analysis, decision to publish, or preparation of the manuscript? The sponsors or funders were not involved in any of these aspects of this research.

- We have amended the financial disclosure statement to include the requested information concerning the role of funders in the research.

We thank the Soprintendenza Archeologica, Belle Arti e Paesaggio per le Province di Lucca and Massa Carrara (Dr. Marta Colombo and Dr. Neva Chiarenza) for their support to the study project. We thank Dr. Enrico Giannichedda and Dr. Monica Baldassarri, who directed the excavations in Aulla (2001-2006) and Montescudaio (2005-2010) respectively. We also thank the Department of Anthropology, the School of Graduate Studies at McMaster University, Mitacs and the Association ‘Amici di San Caprasio’ for their financial support of this research. Thank-you to Dr. Francesco Coschino for his assistance with Fig 1.

AA received finanancial support (Fieldwork and Conference Funding Award, Project ID: 20021030) to conduct this research from the Department of Anthropology, School of Graduate Studies at McMaster University (https://anthropology.mcmaster.ca/) and from Mitacs (https://www.mitacs.ca/our-programs/globalink-research-award/) (Mitacs Globalink Research Award - for research abroad, Project ID: IT30649).

AA, AF and TLP received financial support from the Association "Amici di San Caprasio" to conduct this research.

Did the sponsors or funders play any role in the study design, data collection and analysis, decision to publish, or preparation of the manuscript? No

- Funding Statements have been removed from the Acknowledgement Section.

- We have indicated in our Data Availability Statement that all data are fully available without restriction, and all the available data are already included in the Supplementary files of the manuscript.

- We have included our full ethics statement in the ‘Methods’ section and provided the information requested (lines 309-319).

7. We note that Figure 1 in your submission contain map/satellite images which may be copyrighted. All PLOS content is published under the Creative Commons Attribution License (CC BY 4.0), which means that the manuscript, images, and Supporting Information files will be freely available online, and any third party is permitted to access, download, copy, distribute, and use these materials in any way, even commercially, with proper attribution. For these reasons, we cannot publish previously copyrighted maps or satellite images created using proprietary data, such as Google software (Google Maps, Street View, and Earth). For more information, see our copyright guidelines: http://journals.plos.org/plosone/s/licenses-and-copyright.

We require you to either present written permission from the copyright holder to publish these figures specifically under the CC BY 4.0 license, or remove the figures from your submission:

- We obtained copyright permission from the creator of the map, Dr. Francesco Coschino, to use his map that was created using an open source/copyright free program (© Tuscany Region – GEOscopio, Orthoimagery 2021. Released under Creative Commons Attribution 4.0 International (CC-BY 4.0). Use permitted with attribution to the source.) Please see the attached email permission provided by Dr. Francesco Coschino.

8. We note that there is identifying data in the Supporting Information file. Due to the inclusion of these potentially identifying data, we have removed this file from your file inventory. Prior to sharing human research participant data, authors should consult with an ethics committee to ensure data are shared in accordance with participant consent and all applicable local laws.

-Location data

- The identifying data in the Supporting Information File refers to archaeological human remains from the 11th to 15th centuries CE, with no connection to modern individuals, alive or deceased. The data is presented with codes for each site and burial. There is no identifying information.

- We have added the suggested publications when deemed appropriate.

- Based on the comments of the reviewers we have added six new references (26,27,47,89,119,122), two of these (119,122) were suggested by Reviewer #2. Some references have been moved in the text, and their citation numbers have been changed accordingly in the References.

Reviewer 1:

This is a really well written piece of research with solid questions focusing on infant feeding practices in medieval Tuscany and their possible effects on the sex ratio imbalance. The paper is well structured and arranged, with great historical background, meticulous referencing, as well as proper English.

- Thank you for your feedback.

1. I feel that sometimes, during the train of thoughts, the authors tend to make logical leaps in the discourse, assuming that the reader is aware of the reasoning behind some particular statements. For instance, it is not clearly stated at the beginning for which actual reasons you decided to concentrate on the infant feeding strategies as the main cause of an imbalanced sex ratio in this specific population. Although it is indeed a possible explanation, it wasn't stated why you chose to give priority to this hypothesis rather than others such as nutritional imbalance in adolescence or adulthood, as well as labour or cultural choices. A similar dissonance occurs in the infanticide section, in which it is again unclear how the cited authors arrived to such extreme statements from solely census documents. I would advise to recheck these points, perhaps adding a couple more statements giving more clarity.

- Thank you for this comment. Our rationale for this paper is that childhood is one possible period of life where we might detect discrimination against females through gendered feeding practices. We decided to concentrate our efforts on this particular life stage due to the lack of information about infant feeding strategies during the Middle Ages and because perturbations of infant health can have profound repercussions on adult health. As suggested, we added a better explanation about this choice (lines 85-96).

We agree with the reviewer’s comment about the RS provided by other scholars, with particular focus on infanticide. The paper about infanticide (Coleman, 1971) was based on the assumption that the rich demographic information provided by the polyptych of Saint Germain-des-Pre’s was enough to support the statement that infanticide was the leading cause of skewed sex ratios. The subsequent paragraph (lines 165-172) explains that this hypothesis was already criticized decades ago (e.g., underrepresentation of first-born daughters). Nonetheless, we added extra information about how Coleman (1971) came up with those numbers to help explain the author’s rationale for infanticide (lines 160-161).

2. It is not clear from the practical point of view how you define the weaning process: how and where you see it starting and especially how you see it ending in the isotopic data. You give very specific ages for the end of the process (and not an age range), which give the impression of an abrupt moment while the actual process is quite gradual and often moves back and forth. It would be good to show these clearly in the graphs, perhaps using one as a model, and explain them very clearl

---

## [Editor Report · Decision Letter 1]

26 Nov 2025

Fed the Same Way? Exploring the Influence of Breastfeeding, Weaning, and Childhood Diet on Adult Sex Ratios Through Stable Isotope Analysis of Incremental Dentine in Medieval Tuscany, Italy (11th – 15th c. CE)

PONE-D-25-41106R1

Dear Alessio,

We’re pleased to inform you that your manuscript has been judged scientifically suitable for publication and will be formally accepted for publication once it meets all outstanding technical requirements.

Kind regards,

Dario Piombino-Mascali, Ph.D.

Academic Editor

PLOS ONE
---

## [Editor Report · Acceptance letter]

PONE-D-25-41106R1

PLOS One

Dear Dr. Amaro,

I'm pleased to inform you that your manuscript has been deemed suitable for publication in PLOS One. Congratulations! Your manuscript is now being handed over to our production team.

Kind regards,

on behalf of

Dr. Dario Piombino-Mascali

Academic Editor

PLOS One